# Mechanical Properties Optimization of Hybrid Aramid and Jute Fabrics-Reinforced Graphene Nanoplatelets in Functionalized HDPE Matrix Nanocomposites

**DOI:** 10.3390/polym15112460

**Published:** 2023-05-26

**Authors:** Ulisses Oliveira Costa, Fabio da Costa Garcia Filho, Teresa Gómez-del Río, João Gabriel Passos Rodrigues, Noan Tonini Simonassi, Sergio Neves Monteiro, Lucio Fabio Cassiano Nascimento

**Affiliations:** 1Materials Science Department of Military Institute of Engineering—IME, Rio de Janeiro 22290-270, Brazil; fabiogarciafilho@gmail.com (F.d.C.G.F.); snevesmonteiro@gmail.com (S.N.M.); lucio@ime.eb.br (L.F.C.N.); 2Durability and Mechanical Integrity of Structural Materials Group (DIMME), School of Experimental Sciences and Technology, Rey Juan Carlos University, C/Tulipán, s/n. Móstoles, 28933 Madrid, Spain; mariateresa.gomez@urjc.es; 3Catalysis Laboratory for Polymerization, Recycling and Biodegradable Polymers (LCPRB), Professor Eloisa Mano Macromolecules Institute—IMA, Rio de Janeiro 21941-598, Brazil; jgprodrigues@ima.ufrj.br; 4Materials Science Department of State University of Northern Rio de Janeiro—UENF, Campos dos Goytacazes, Rio de Janeiro 28013-602, Brazil; noantoninisimonassi@gmail.com

**Keywords:** aramid fabric, jute fabric, graphene nanoplatelet, high-density polyethylene, hybrid nanocomposite

## Abstract

Natural lignocellulosic fibers (NLFs) have been used as a reinforcement for polymer matrix composites in the past couple of decades. Their biodegradability, renewability, and abundance make them appealing for sustainable materials. However, synthetic fibers surpass NLFs in mechanical and thermal properties. Combining these fibers as a hybrid reinforcement in polymeric materials shows promise for multifunctional materials and structures. Functionalizing these composites with graphene-based materials could lead to superior properties. This research optimized the tensile and impact resistance of a jute/aramid/HDPE hybrid nanocomposite by the addition of graphene nanoplatelets (GNP). The hybrid structure with 10 jute/10 aramid layers and 0.10 wt.% GNP exhibited a 2433% increase in mechanical toughness, a 591% increase in tensile strength, and a 462% reduction in ductility compared to neat jute/HDPE composites. A SEM analysis revealed the influence of GNP nano-functionalization on the failure mechanisms of these hybrid nanocomposites.

## 1. Introduction

The 21st century has brought increased attention to environmental sustainability, leading to a shift away from synthetic materials and towards natural alternatives [1,2]. One of these alternatives led to the rise of natural lignocellulosic fibers (NLFs) as a reinforcement of polymer matrix composites. Indeed, NLFs have gained popularity in various engineering sectors, including civil construction [3,4,5], automotive [6,7], packing [8,9,10], and ballistic armor [11,12,13,14,15]. This was especially possible owing to their advantages over synthetic fibers. These advantages include lower density, biodegradability, abundant availability, good damping properties, machinability, high health safety, ease of separation for recycling and enhanced energy recovery, as well as CO_2_ neutrality [2]. Despite such advantages, the production of NLFs may be impaired by a lack of quality control, as the weather, time of harvest, or soil composition could lead to heterogeneous properties [16]. To address the reliability of NLF composites, hybrid composite structures, combining both natural and synthetic fibers, are being developed to produce materials with superior mechanical performance, durability, and other desirable characteristics [17,18,19,20].

While most NLFs such as flax, hemp, or jute are cost-effective, low in density, and environmentally friendly, they may have limitations in terms of their strength, stiffness, as well as resistance to moisture and UV radiation. Synthetic fibers, on the other hand, exhibit exceptional strength, stiffness, and resistance to heat, chemicals, and abrasion. Nonetheless, their manufacturing cost could be way more expensive. Therefore, by blending these fibers in hybrid composite structures, it is possible to produce materials with improved mechanical properties. High strength and stiffness, while maintaining a low density and environmental friendliness as well as more cost-effective production could be achieved for such hybrid structures [21]. Aramid fibers are a typical example of a synthetic component used in these composites due to their exceptional characteristics [22].

A recent trend aiming to further improve the properties of these hybrid composites lies on the use of a graphene-based material. Graphene, a two-dimensional (2D) nanomaterial with exceptional mechanical, thermal, and electrical properties, could be applied in these hybrid composite structures. Indeed, graphene-filled NLFs and/or synthetic fiber composites have shown promise in improving strength, stiffness, and electrical conductivity, making them suitable for aerospace and defense applications [23,24,25,26,27,28,29]. Jesuarockiam et al. [30] investigated the effect of the addition of graphene nano platelets (GNP) in the thermal and dynamic mechanical properties of a Kevlar/Cocos nucifera natural fiber reinforcing epoxy matrix composite. They verified that the addition of GNP resulted in a higher thermal stability of the hybrid composite as well as enhanced viscoelastic properties which were due to the thermal barrier and effective crosslinking exhibited by the addition of 0.75 wt.% of GNP. Similarly, Oun et al. [31] shown that the incorporation of graphene in hybrid composite structures resulted in an increase in the interlaminar shear strength properties of the composites at an in-service elevated temperature. They suggested that the addition of graphene improved the bonding mechanism between different fibers and filler which modified the main failure mechanism from pull-out failure at low temperatures to delamination failure at elevated temperatures. Kishore et al. [32] showed that the hybridization of a jute/basalt reinforced epoxy composite by the addition of graphene in the range of 0.20 up to 0.60 wt.% led to an increase in the surface roughness of the composite. The combination of these materials in a polymer matrix is an exciting area for creating high-performance and sustainable materials. In this context, the present work investigated the development of hybrid composites of a jute and aramid fiber-reinforced high-density polyethylene (HDPE) matrix with functionalization by GNP. These novel hybrid nanocomposites present a promising performance for engineering applications.

## 2. Materials and Methods

### 2.1. Jute Fabric (J)

Among the numerous fiber-reinforced natural composites with potential engineering applications, jute (*Corchorus capsularis*) stands out as a remarkable choice. It is globally recognized and extensively studied, making it one of the most well-known natural fibers available [1,2]. Not only is jute cost-effective, but it also boasts exceptional strength among other natural fibers. Traditionally, jute fiber has been utilized in the manufacturing of bags, carpets, yarn, and ropes. Additionally, industries such as automotive, construction, and packaging have embraced jute fiber as a reinforcing material [33].

Jute is primarily cultivated in humid and tropical regions, particularly in northern Brazil [33]. What sets jute apart is that it is a vegetable textile fiber originating from the *Tilioideae* family. The jute plant typically grows to a height of 3 to 4 m, with a stalk diameter of approximately 20 mm, as depicted. The valuable fiber lies between the bark and the inner stalk, and its extraction is accomplished through the maceration process. Therefore, jute fabric was obtained from the Brazilian manufacturer Sisalsul. According to Monteiro et al. [33], the fiber density was considered as 1.30 g/cm^3^. This parameter was important for estimating the fiber volume fraction. The as-received jute fabric consisted of a simple weave and was cut to a dimension of 120 × 120 mm and carefully dried in an oven at 60 °C for 24 h to remove the moisture.

### 2.2. High-Density Polyethylene (HDPE)

Pellets of high-density polyethylene (HDPE) grade HE150 were acquired from Braskem in São Paulo, Brazil. According to the product specifications, the density of HDPE is 0.948 g/cm^3^. The HE150 resin was created specifically for the monofilament extrusion process. It is commonly used for the extrusion of oriented structures because it has a low gel level and a great balance of processability, dependability, and stability [34].

### 2.3. Graphene Nanoplatelets (GNPs)

The GNP used in this study was supplied by UCSGraphene in Caxias do Sul, Brazil. It was in powder form, consisting of nanoparticles with 10 to 50 layers of graphene. The powder formed agglomerates with a lateral size of up to 25 μm, as shown in Figure 1, which were used to decorate the HDPE matrix in the nanocomposites. The amount of GNP filler utilized to the hybrid nanocomposites was 0.10 wt.%. Based on previous studies, it was shown that 0.10 wt.% GNP improves the mechanical and thermal properties of the polymeric matrix [35].

### 2.4. Aramid Fabric (A)

Aramid fabric, commercially named Twaron^®^, was provided by Teijin Aramid, São Paulo, Brazil. The commercial grade used in the present work was the 410 g/m^2^ ballistic fabric, CT 736, with basket 2 × 2 weave. This aramid fabric is indicated for the production of modern, state-of-the-art ballistic helmets typically used in mine boots sandwich constructions, having good processability with various resin systems [36]. The Twaron^®^ fabric was cut to 120 × 120 mm, in order to fabricate the aramid reinforced composites.

### 2.5. Nanocomposites Fabrication

First, the GNP powder was mechanically agitated with HDPE to create a concentrate. The concentrate, consisting of a mixture of GNP and HDPE pellets, was then diluted to a weight fraction of 0.10 wt.% using an interpenetrating, co-rotating, twin-screw extruder Tecktril (model DCT-2). The extrusion conditions were set according to Escocio et al. [37] as follows: a screw rotation of 300 rpm, feeder rotation of 15 rpm, and temperature settings in the processing zones as follows: (i) first zone at 90 °C; (ii) second to fifth zones at 140 °C; and (iii) sixth to ninth zones at 160 and 180 °C.

Next, 300 µm-thick films were produced from the GNP functionalized HDPE (GNP/HDPE) pellets through hot compression molding at 150 °C using a heat press. As depicted in Figure 2, a laminate pattern was employed to create the hybrid nanocomposite plates, with alternating layers of fabric and polymeric films. To achieve a 50 vol.% of reinforcement in the polymeric matrix, based on the methodology of Tomasi Tessari et al. [38], 20 layers of jute fabric and 21 layers of GNP/HDPE films were utilized.

The processing involved gradually increasing the pressure by one ton per min for each new pressure step, with 30 s of degassing. This sequence was repeated until a pressure of 13 tons was reached. Subsequently, the hybrid nanocomposite plates, referred to as jute/aramid/GNP/HDPE, were cooled down to room temperature (RT). Each plate had dimensions of 120 × 120 × 10 mm. To determine the density of the hybrid nanocomposite plates, an Archimedes test was conducted, and the results were further validated through geometric measurements, yielding a density of 0.92 ± 0.031 g/cm^3^.

### 2.6. Hybrid Aramid and Jute Fabrics-Reinforced GNP/HDPE Nanocomposites

Regarding the jute/aramid/GNP/HDPE hybrid nanocomposites, the optimal number of layers for different synthetic and natural fiber reinforcements is a critical factor in the design of hybrid nanocomposites. In this study, a total of 20 layers of both jute and aramid fabrics were utilized for each hybrid nanocomposite, corresponding to a total of 50 vol% of fabrics. To optimize cost-effectiveness, a strategic approach was adopted based on the sequence depicted in Figure 3, involving variations in the aramid and jute layers. Table 1 presents the configuration and associated nomenclature of nanocomposites, both hybrid- and single-reinforced, with a GNP-functionalized HDPE matrix. Each layer corresponds to a fraction of 2.5 vol% in the nanocomposite.

For clarity to the reader, along the text, nanocomposites refer to the GNP-functionalized HDPE matrix and single nanocomposites to those without either aramid or jute fabrics.

### 2.7. Tensile Test

The tensile tests were performed at RT using an INSTRON 3365 universal machine. The test speed and cell load parameters were 2 mm/min and 10 KN, respectively. Seven samples were cut manually with a bandsaw to the dimensions of the samples 120 × 15 × 10 mm, adapted from the ASTM D3039 standard [39]. From the tensile test, the elastic modulus (E), tensile strength ($\sigma_{u}$), and ductility (ε) were calculated for all the composites. These parameters provide valuable information regarding the mechanical properties of the composites and their ability to withstand tensile forces.

### 2.8. Scanning Electron Microscopy (SEM)

Microscopic analyses were performed in order to observe the fracture surface of the composites after the tensile tests, using a scanning electron microscope (SEM Quanta FEG 250, FEI), Rio de Janeiro, Brazil, operating with secondary electrons accelerated at 20 kV. The samples were sputter coated with gold in a LEICA equipment model EM ACE600. This allowed us to observe and analyze the characteristics and patterns of the fracture, providing valuable insights into the structural behavior and failure mechanisms of the composites under tensile and impact loadings.

### 2.9. Raman Spectroscopy

Raman spectroscopy was carried out for the GNPs, pure HDPE, and 0.10 wt.% GNP-functionalized HDPE films. Raman spectra were obtained with RT backscatter geometry using a Raman spectrometer equipped with an Andor Shamrock spectrometer with an iDus Charge Coupled Device (CCD) detector, a 488 nm (~2.54 eV) laser, and an optical system. All measurements were obtained using a laser spot diameter of 1 μm and power of 1 mW. The spectral broadening of the spectrometer for this configuration was determined using a silicon wafer peak at 520 cm^−1^ fitted using a Gaussian line shape with a maximum half width (FWHM) of 4 cm^−1^. Raman spectra were obtained using the Origin Pro data analysis software.

### 2.10. DSC Analysis of the HDPE and GNP/HDPE Nanocomposites

The DSC analysis of the nanocomposites was conducted using an aluminum crucible in a TA Instruments calorimeter model Q1000. The equipment operated in a nitrogen atmosphere with heating rates of 10 °C/min, within a temperature range of 20 to 200 °C. Through an exothermic and endothermic events analysis of heat flux versus temperature curves, the crystallization (T_c_) and melting (T_m_) temperatures of the GNP/HDPE nanocomposites were determined, along with the degree of crystallinity using Equation (1) based on the work of Evgin et al. [40].
(1)$$X_{c}=\frac{{∆H}_{m}}{\left[{∆H}_{0}\times \left(1-\emptyset \right)\right]}\times 100$$
where Δ*H* represents the melting heat of the sample, Δ*H*_0_ = 293 J/g refers to the melting heat for 100% crystalline HDPE [41], and ∅ denotes the weight fraction of the nanofiller in each sample. Furthermore, a non-isothermal crystallization kinetics analysis was conducted on the HDPE and GNP/HDPE nanocomposites using cooling rates of 5, 10, 15, and 20 °C/min. It was crucial to assess the influence of the GNP on the crystallization process of the GNP/HDPE matrix. Furthermore, the study delved into the impact of GNP on nucleation and crystal growth mechanisms. These findings provide valuable insights into the behavior of the nanocomposites and contribute to a better understanding of their properties.

### 2.11. Izod Impact Tests

The Izod impact test was conducted using a Pantec CHIZ-25, 220 V × 60 Hz pendulum with a 22 J hammer. Specimens were cut and machined from composite plates according to the dimensions specified by the ASTM D256 standard [42]. For each treatment, five samples were produced, each measuring 62.5 × 12.7 × 10 mm. This comparative analysis was crucial for evaluating and identifying the most enhanced material among all the nanocomposites, including the hybrid nanocomposites. Such findings are instrumental in determining the optimal material for engineering applications.

### 2.12. Statistical Analysis

The analysis of variance (ANOVA) was employed using the F test to determine if there were any statistically significant differences between the mean values of the results obtained from Izod impact and tensile tests. A confidence level of 95% was utilized for all the tests, as previously reported [43,44].

Once the presence of significant differences among the mean values of the results for different composite treatments was established, the Tukey test, also known as the honestly significant difference (HSD) test, was employed. The objective was to statistically evaluate the impact of GNP in the HDPE matrix, as well as the number of aramid layers in the hybrid nanocomposites. The Tukey test is a hypothesis test that involves rejecting the null hypothesis of equality based on the Minimum Significant Difference (*msd*), as calculated by:(2)$$msd=q\times \sqrt{\frac{MSE}{r}}$$
where *q* is the total amplitude studied, which is a function of the degree of freedom (DF) of the residue and the number of treatments; *MSE* is the mean square error; and r is the number of replicates of each treatment [44].

## 3. Results and Discussion

### 3.1. Raman Spectroscopy

The Raman spectra index of all bands found are disclosed in Appendix A. Based on Figure 4a, the bands between 1400 and 1480 cm^−1^ in the spectral region from 200 to 4000 cm^−1^ correspond to methylene bending vibrations (𝛿(CH_2_)). Additionally, in Figure 4b, it can be observed that the band at 1440.00 cm^−1^ refers to the D band present in the nanoplatelets, but with a displacement compared to the same band when analyzed separately, as discussed in Figure 5. From the GNP/HDPE spectrum, one can notice a small band in 1580.80 cm^−1^ attributed to the D band of the GNP. In addition, the crystalline phase is represented by the band at 1418.70 cm^−1^, known as the crystallinity band [45,46,47,48]. This crystallinity band at 1418.70 cm^−1^ is commonly used to determine the degree of orthorhombic crystallinity of the HDPE [45]. The bands at 1060.00 and 1367.30 cm^−1^ are attributed to the amorphous phase, while the Raman bands at 1060.80 and 1127.70 cm^−1^ are attributed to the symmetric and asymmetric stretching vibrations of the C–C bonds. The Raman bands at 2849.70 cm^−1^ and 2883.80 cm^−1^ are attributed to the stretching of groups (CH_3_), and the bands at 2907.70 cm^−1^ and 3068.50 cm^−1^ are assigned to the (C–H) group [45,46,47,48].

Based on Figure 5, the Raman spectrum of the GNP revealed the presence of the G band at 1573.6 cm^−1^ and the D band at 1347.7 cm^−1^. The occurrence of the G band is attributed to the first-order scattering of the *E*_2*g*_ mode, while the D band is associated with defects in the graphite lattice, as explained by Krishnamoorthy et al. [49]. Moreover, besides the changes observed in the G and D bands in the graphene sheets, significant alterations were also observed in the overtone band of graphene, which normally occurs at higher wavenumbers. In the case of the GNPs, 2D overtones were observed at 2715.9 cm^−1^ [49].

### 3.2. DSC Analysis

Differential scanning calorimetry (DSC) analyses were conducted on the HDPE and GNP/HDPE nanocomposites with 0.10 wt.% of GNP and different heating and cooling rates (5–20 °C/min). The results, presented in Table 2, revealed the crystallization (T_c_) and melting temperatures (T_m_). Notably, with the incorporation of the GNP into the HDPE matrix, both T_c_ and T_m_ showed an average shift of 1 °C towards higher temperatures compared to pure HDPE. This observation supports the nucleation effect caused by the presence of graphene in the polymeric matrix, as emphasized by Evgin et al. [40].

Furthermore, the crystallization behavior of the GNP/HDPE nanocomposites is influenced by the agglomeration of graphene nanoparticles, which may explain the observed increase in crystallization temperature with the increasing graphene content. The clustering of these nanoparticles can hinder the regular packing of HDPE polymeric chains, as pointed out by Evgin et al. [40], resulting in reduced HDPE crystallinity despite acting as nucleating agents.

Moreover, it was observed that the crystallization and melting temperatures of the nanocomposites were slightly higher than those of pure HDPE, particularly at lower cooling rates. This can be attributed to the time-dependent nature of crystallization and melting processes, where lower cooling rates allow for increased fluidity and diffusivity of molecules due to lower viscosity and more time available for the completion of crystallization and melting. Consequently, both Tc and Tm temperatures increase with the decreasing cooling rate, as reported by Evgin et al. [40] as well as Jiang and Drzal [50]. These findings indicate that the amount of added GNPs plays a crucial role in the overall structure and properties of the composite.

In Figure 6, the crystallization temperature of the HDPE and nanocomposites is plotted against the cooling rate. These results were obtained by analyzing the exothermic peaks in the DSC cooling curves, as depicted in Appendix A, with a heating rate of 10 °C/min. It can be observed that the crystallization peak temperature of the GNP/HDPE nanocomposites is slightly higher than that of pure HDPE for all cooling rates. This suggests that the presence of GNP particles in the HDPE matrix can induce heterogeneous nucleation by creating nucleation sites for the polymeric chains during melt crystallization, particularly at rates of 10 and 15 °C/min, where this effect is intensified.

Interestingly, at low cooling rates, the incorporation of GNP does not seem to significantly influence the crystallization temperature of the nanocomposites. However, for rates greater than 10 °C/min, crystallization temperatures show a slight increase with an increasing percentage of GNP in the HDPE matrix. These results suggest that the effect of GNP on crystallization temperature is more prominent at higher cooling rates, indicating a rate-dependent influence of GNP on the crystallization behavior of the nanocomposites.

As a result, from the DSC heating curves at a rate of 10 °C/min for the HDPE and GNP/HDPE films, shown in Appendix A, the degree of crystallinity for all nanocomposites was calculated using the melting enthalpy from the endothermic peak, applying Equation (1). The resulting values are as follows: HDPE (46.10%) and GNP/HDPE (62.19%).

Furthermore, in terms of crystallization kinetics, the heat of crystallization can be converted into the relative degree of crystallinity (*X_T_*) by dividing the heat released at each crystallization temperature *T* (Δ*H_Τ_*) by the total heat (Δ*H*_0_) corresponding to complete crystallization [51]:(3)$$X_{t}=\frac{\int_{T_{0}}^{T_{c}}\left(\frac{dH}{dT}\right)dT}{\int_{T_{0}}^{T_{\infty }}\left(\frac{dH}{dT}\right)dT}$$
where $T_{0}$, $T_{c}$, and $T_{\infty }$ represent the initial and final temperatures of crystallization at time *t* and the final temperature of crystallization, respectively, while dH represents the enthalpy of crystallization released during an infinitesimal temperature interval dT: the crystallization time (t) can be obtained from the crystallization temperature using the following equation:(4)$$t=\frac{T_{0}-T}{\varphi }$$
where $T_{0}$ and T represent the temperature at the beginning and time of crystallization (t), respectively, and *φ* is the cooling rate. The relative degree of crystallinity (*X_T_*) is depicted in Figure 7a,b as a function of temperature (T). Additionally, Figure 7c,d illustrate the relative degree of crystallinity (*X_T_*) as a function of time (t). These curves clearly exhibit a strong dependence of the crystallization process on the cooling rate. As crystallization progresses, the curves tend to approach a plateau due to the impact and crowding of the spherulites [51]. Moreover, higher cooling rates result in shorter times required for crystallization to occur.

The time from the initiation of crystallization to the point where the relative degree of crystallinity reaches 50% was calculated from the curves in Figure 7 and plotted for each cooling rate *φ*, as illustrated in Figure 8. It can be observed that the t_1/2_ value of the GNP/HDPE nanocomposites is slightly higher compared to pure HDPE at a given cooling rate, particularly noticeable at lower cooling rates. This suggests that the addition of graphene filler complicates the time required for the polymer to achieve half the degree of crystallization, resulting in a lower rate of crystallization due to the effect of heterogeneous nucleation. Moreover, it appears that the charge load does not have a significant impact on the t_1/2_ value in the GNP/HDPE nanocomposites.

In general, the kinetics of the crystallization transition has been traditionally interpreted using the Avrami and Ozawa equations [40,50,51]. However, it has been noted in the literature that these methods may not fit well to non-isothermal crystallization data. Therefore, to better describe the non-isothermal crystallization process, Liu et al. [52] proposed a combination of the Avrami and Ozawa equations:(5)$$\mathrm{Log}\left(k\right)+nlog\left(t\right)=LogK\left(T\right)-mlog\left(\varphi \right)$$
(6)$$log\left(\varphi \right)=LogF\left(t\right)+\alpha Log\left(t\right)$$

The parameter *F*(*T*), defined as *F*(*T*) = [*K*(*T*)/*Z_t_*]^1/*m*^, where *φ* is the cooling rate and α is the ratio between the exponents Avrami and Ozawa n/m, carries a significant physical and practical meaning. It represents the critical cooling rate required to achieve a specific degree of crystallinity within a unit of crystallization time.

Figure 9a,b displays plots of *Log*(*φ*) versus *Log*(*t*) at various degrees of crystallinity for the HDPE and GNP/HDPE nanocomposites, respectively. Notably, these graphs exhibit excellent linearity, with most coefficient of determination (R-squared) values exceeding 0.95 in all cases, as one can notice in Table 3.

The values of the α and Log[F(T)] parameters were determined from the intercepts and slopes of the fitted lines, respectively, as presented in Table 3. It was observed that the *Log*[*F*(*T*)] value increases with an increasing degree of crystallinity. This can be attributed to the fact that at low values of *X_T_*, the polymeric matrix is in a molten state, resulting in a higher crystallization rate.

Furthermore, the *Log*[*F*(*T*)] parameter for the GNP/HDPE nanocomposites exhibits slightly higher values compared to pure HDPE. This suggests that the presence of GNP particles did not facilitate the crystallization process in the HDPE composites, resulting in lower crystallization rates at higher crystalline fractions [51,53].

The *α* values remain parallel and nearly constant for a given composition and different *X_T_* values. Additionally, the ratio of the Avrami exponent to the Ozawa exponent ranges from 1.8 to 2.0, which is consistent with findings in the literature [51,53]. This suggests that significant growth of secondary crystallization accompanies primary crystallization during the non-isothermal crystallization period. The nucleation mechanism and crystal growth geometries are similar, indicating that the proposed method successfully describes the non-isothermal crystallization process of HDPE and its nanocomposites.

Furthermore, Friedman’s differential isoconversional method [54] was employed to evaluate the effective activation energies, as shown:(7)$$Ln{\left(\frac{dX_{T}}{dt}\right)}_{X_{T}}=C-\frac{{∆E}_{X_{T}}}{{RT}_{X_{T}}}$$
where $\frac{dX_{T}}{dt}$ is the instantaneous crystallization rate as a function of time for a given value of relative crystallinity ($X_{T}$), R is the universal gas constant, and ${∆E}_{X_{T}}$ is the effective energy barrier of the process for a given value of $X_{T}$. This approach requires performing a series of experiments in different temperature programs to obtain this energy. Specifically, at various cooling rates, the values of $\frac{dX_{T}}{dt}$ at a specific $X_{T}$ are correlated with the corresponding crystallization temperature at this $X_{T}$, i.e., $T_{X_{T}}$, a straight line, can be obtained by plotting $Ln{\left(\frac{dX_{T}}{dt}\right)}_{X_{T}}$ versus 1/$T_{X_{T}}$, and the slope is −$\frac{{∆E}_{X_{T}}}{R}$. The dependence of such an energy barrier on the relative crystallinity for HDPE and its GNP/HDPE nanocomposites based on the Friedman equation is shown in Figure 10.

An analysis of Figure 10 reveals that the GNP/HDPE nanocomposite exhibited lower activation energies for low crystallinity layers $\left(X_{T}\leq 20\%\right)$, indicating a pronounced nucleating effect of GNP on the HDPE matrix, facilitating the initial crystallization process. However, for larger transformed layers $\left(X_{T}>20\%\right)$, graphene was found to cause an increase in activation energy. This could be attributed to the need to expel the GNP during crystal growth, resulting in a more complex and challenging process. This observation is consistent with the findings in Figure 8, further supporting the conclusion.

### 3.3. Tensile Properties

Through the tensile tests, the tensile strength (σ_u_), elastic modulus (E), ductility (ε_t_) (deformation at break), and toughness (T) (area under the curve σ vs. ε) were obtained. Figure 11 shows the results for the E of the 20J/HDPE and 20J/GNP/HDPE single nanocomposites.

The incorporation of nanoload is accompanied by an observed increase in the value of E, as depicted in Figure 11. Thus, in terms of cost-effectiveness, smaller amounts of GNP are found to be more efficient in improving the mechanical behavior. This phenomenon may be associated with a points of stress concentration caused by the agglomeration of GNPs in the HDPE matrix, as noted in previous studies [35,55,56].

However, the values of E obtained for the 20J/HDPE composite (1.32 ± 0.73 GPa) in this study are considerably higher than those reported in the literature (~1.0 GPa) [14,57,58]. An increase of over 20% was demonstrated by the 20J/GNP/HDPE single nanocomposite (1.63 ± 0.15 GPa) compared to the 20J/HDPE studied in this work. Furthermore, when compared to the value reported in the literature, the increase exceeds 60% [14,57,58].

As for tensile strength, Figure 12 presents the average values and their standard deviations. Due to overlapping standard deviation bars, ANOVA and Tukey tests were conducted to ascertain whether the mean values exhibit any significant differences.

Based on the results of Figure 12, it can be concluded that the mean values of tensile strength are equal. This suggests that the incorporation of GNPs did not significantly alter the tensile strength of the HDPE matrix. It can also be inferred that a lower nanoload content of GNPs performs better compared to higher amounts, likely due to the agglomeration and alignment of GNPs in the HDPE matrix, as reported in previous studies [35,59,60,61,62].

Similarly, the results indicate that the inclusion of GNPs in the HDPE matrix did not have a significant influence on the ductility (ε_t_) of the nanocomposite, 4.71%, as shown in Figure 13. The lack of influence on ductility can be attributed to the low compatibility between GNPs and the HDPE matrix, resulting in voids around GNPs and their agglomeration in the HDPE matrix after plastic deformation, as reported in previous studies [62]. These characteristics lead to a decrease in the surface area of GNPs, reducing the load transmission through the matrix, and maintaining the ductility of the polymer similar to that of pure HDPE.

The data provided in Figure 14 show that toughness, which is the energy required to cause fractures in a composite material, can be calculated from the stress–strain curve or from the tensile test data sheet. The data compare different nanocomposites made of HDPE, jute, and GNP.

The 20J/HDPE composite presented a toughness value of 0.66 MJ/m^3^, as shown in Figure 14. However, the incorporation of GNP into the HDPE matrix improves the toughness of the single nanocomposite 20J/GNP/HDPE to 0.93 MJ/m^3^, which represents a more than 40% increase compared to 20J/HDPE. This indicates that the addition of GNP to the nanocomposite makes it more resistant to fracture. The increase in toughness can be attributed to the reinforcing effect of GNP, which improves the overall mechanical properties of the nanocomposite.

Figure 15a illustrates the typical shear band morphology and fracture characteristics of the 20J/HDPE composites. The fracture mechanism of the 20/J/HDPE composites differs from that of the 20J/GNP/HDPE nanocomposites, as evident in Figure 15b. This fibrillation is observed as part of the tearing process due to significant localized plastic deformation. SEM micrographs of nanocomposite materials containing 0.10 wt.% of GNP show the presence of voids and the initiation of void cracking, with polymeric fibrils stabilizing the voids [63].

The images in Figure 15 reveal that the GNP/HDPE matrix deforms between the fibers of the jute fabric, indicating a strong adhesion of the HDPE and GNP/HDPE matrices to the jute fabric, filling the gaps between the fabric layers. SEM analysis confirms that shear is the mechanism governing the ductile fracture in the 20J/HDPE composites, while fibrillation and crazing fracture are the mechanisms for the 20J/GNP/HDPE nanocomposites with larger diameter sizes and GNP content [63].

For the composites in group 2, a GNP concentration of 0.10 wt.% was chosen as it optimizes the tensile properties of all the composites in group 1, while minimizing the cost increase and reducing the likelihood of agglomerate formation and internal defects. Therefore, the tensile results for the composites in group 2 with 0.10 wt.% GNP are discussed below. Figure 16 presents the tensile strength results corresponding to the ultimate stress reached by the single and hybrid nanocomposites. It is observed that as the number of aramid layers increases, the tensile strength tends to increase, indicating a better compatibility of the HDPE matrix with synthetic aramid fibers compared to natural fibers, such as jute.

Furthermore, there is a significant difference in the contribution to tensile strength between synthetic and natural fibers, with aramid fibers being approximately five times stronger than jute fibers [33]. Based on these results, the ANOVA was conducted to verify the significant differences with 95% confidence, and the Tukey test was used to identify the differences between composites when compared pairwise.

The results of the ANOVA indicated that the calculated F-value of 254.09 was significantly higher than the critical F-value of 3.06, indicating that the values were not equal. The Tukey test was then conducted with a value of msd = 47.41, and the results are presented in Table 4. It can be observed from the table that all the values showed significant differences in relation to each other. Furthermore, it can be noted that as the number of layers of aramid fabric increased, the differences between the averages also increased. Specifically, when compared to the 20J/GNP/HDPE single nanocomposite, the 15J/5A/GNP/HDPE, 10J/10A/GNP/HDPE, 5J/15A/GNP/HDPE, and 20A/GNP/HDPE hybrid nanocomposites exhibited differences in tensile strength of 195%, 591%, 1411%, and 1622%, respectively.

Figure 17 presents the average values for the E of single and hybrid nanocomposites with varying layers of jute and aramid fabrics. It can be observed that nanocomposites with higher layers of jute fabric tend to have relatively constant and lower values of E compared to composites with higher layers of aramid fabric. The results of the ANOVA indicated that the calculated F-value of 6.36 was greater than the critical F-value of 3.06, with a 95% reliability. This suggests that the incorporation of aramid fibers significantly influences the increase in the value of E of the hybrid nanocomposites.

Based on the Tukey test results, with an msd equal to 501.04 as shown in Table 5, the 20A/GNP/HDPE single nanocomposite exhibited the most significant difference compared to the other composites. Specifically, in comparison to the 20J/GNP/HDPE single nanocomposite, the 20A/GNP/HDPE single nanocomposite demonstrated an increase of nearly 40% in the modulus of elasticity, suggesting better compatibility between the synthetic fibers and the HDPE matrix. However, for nanocomposites with layers below 50% of both synthetic and natural fibers, delamination was observed at the interface between the different fibers, resulting in an E similar to that of the 20J/GNP/HDPE single nanocomposite.

Figure 18 presents the mechanical toughness modulus results of the single and hybrid nanocomposites with varying layers of synthetic and natural fibers. This parameter is determined by calculating the area under the stress versus strain curve or by using the values provided by the equipment’s data. It represents the energy required to fracture the material under applied tensile stresses in a quasi-static manner. Notably, a significant increase in energy is observed with a higher incorporation of synthetic fibers, indicating the substantial improvement in mechanical properties achieved by using synthetic fibers in the hybrid nanocomposites.

Based on the ANOVA results, the calculated F value of 133.28 is greater than the critical F value of 3.06, indicating that the averages of the mechanical toughness values are not equal with 95% reliability. Further, the Tukey test with msd = 6.77, as shown in Table 6, reveals that all hybrid nanocomposites with 10 layers of aramid fabric exhibit significant differences in mechanical toughness, with higher numbers of aramid layers promoting higher values of toughness. This underscores the potential of using different fibers in polymer matrix hybrid nanocomposites.

However, upon closer examination of Table 6, it can be observed that the hybrid and single nanocomposites with 15 and 20 layers, respectively, of aramid fabric show similar mechanical toughness values. This suggests that both the hybrid nanocomposites 5J/15A/GNP/HDPE and 10J/10A/GNP/HDPE may be considered the most optimized in terms of mechanical toughness.

Furthermore, the 5J/15A/GNP/HDPE hybrid nanocomposite (37.92 ± 5.33 MJ/m^3^) exhibited a remarkable increase of 3977% compared to the 20J/GNP/HDPE single nanocomposite (0.93 ± 0.09 MJ/m^3^). Notably, the 10J/10A/GNP/HDPE hybrid nanocomposite (23.56 ± 1.60 MJ/m^3^) showed an impressive increase of 2433% in relation to the 20J/GNP/HDPE single nanocomposite in terms of mechanical toughness.

In terms of ductility, as shown in Figure 19, defined by the maximum deformation achieved by the single and hybrid nanocomposites, a significant increase in maximum deformation is observed with increasing layers of aramid fabrics in the hybrid nanocomposite. For instance, the 10J/10A/GNP/HDPE hybrid nanocomposite exhibited a 462% increase in maximum deformation (26.47%) compared to the 20J/GNP/HDPE single nanocomposite (4.71%). This result underscores the potential of combining synthetic and natural fibers in the thermoplastic matrix, which could lead to the production of ballistic helmets with improved cost-effectiveness and superior mechanical properties, a critical aspect in helmet design.

The ANOVA results revealed that the calculated F value of 62.68 exceeded the critical F value of 3.06, indicating that the means are not equal with 95% reliability. Further analysis using the Tukey test with an msd value of 5.13 helped to identify the treatments that showed significant differences. Upon examining Table 7, it is evident that all nanocomposites exhibited statistically significant differences from each other, except for those with a number of aramid layers greater than 10.

In Figure 20, the fracture mechanisms of the hybrid nanocomposites are illustrated. The fracture surface indicates notable tearing and fibrillation of the HDPE, as well as delamination of the aramid and jute fabric. The insert in Figure 20 also reveals that the GNP/HDPE matrix remained attached to the aramid fabric post the tensile test. This observation suggests that there is a harmonious relationship between the GNP/HDPE and the synthetic fabric.
Figure 20Fracture surface of the hybrid nanocomposites.
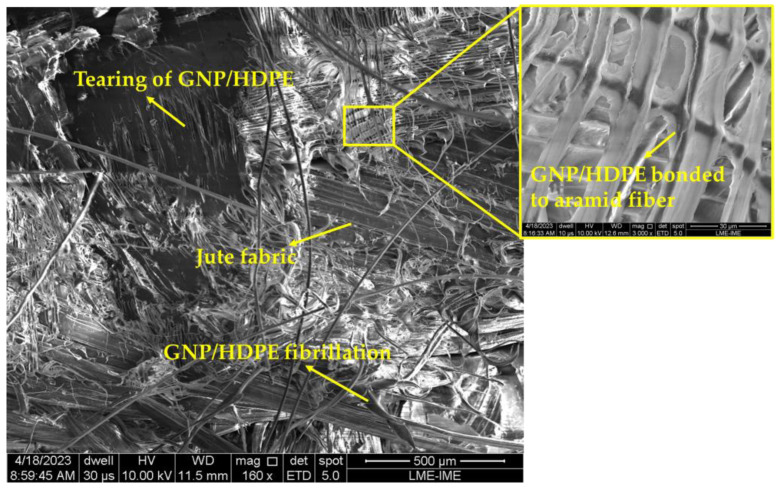

polymers-15-02460-t008_Table 8Table 8Summary of the tensile properties discussed in the present work in comparison with other works in the literature.
Tensile Strength (MPa)Elastic Modulus (GPa)Ductility (%)Toughness (MJ/m^3^)Ref.20J/HDPE25.53 ± 3.691.32 ± 0.734.67 ± 0.510.66 ± 0.11PW20J/GNP/HDPE24.70 ± 0.721.63 ± 0.154.71 ± 0.750.93 ± 0.0915J/5A/GNP/HDPE72.77 ± 9.741.66 ± 0.2214.83 ± 1.616.42 ± 0.8910J/10A/GNP/HDPE170.78 ± 30.601.60 ± 0.1226.47 ± 3.2223.56 ± 1.605J/15A/GNP/HDPE373.26 ± 8.941.85 ± 0.2723.75 ± 3.4337.92 ± 5.3320A/GNP/HDPE425.42 ± 45.042.25 ± 0.3924.80 ± 2.8238.98 ± 5.80Curaua/Epoxy134.67 ± 23.013.08 ± 0.557.87 ± 1.175.83 ± 1.61[64]Curaua/0.10%GO/Epoxy190.20 ± 31.754.95 ± 0.476.94 ± 0.907.50 ± 1.92[64]Curaua non-woven/Epoxy44.48 ± 13.113.87 ± 0.432.71 ± 0.380.66 ± 0.25[65]19Aramid/Epoxy280.21 ± 53.7911.40 ± 1.933.97 ± 0.305.41 ± 1.25[65]2Curaua/10Aramid/Epoxy156.13 ± 9.937.26 ± 1.424.01 ± 0.383.16 ± 0.50[65]2.0%PEG/SiO_2_/HDPE27.00 ± 1.500.80 ± 0.0545.00 ± 5.00-[66]HDPE17.00 ± 2.000.45 ± 0.02550.00 ± 4.00-[67]PW—Present work.


Table 8 provides a summary of the properties discussed earlier for the aforementioned single and hybrid nanocomposites, as well as for some composites reported in the literature with thermoset matrices and pure HDPE. Notably, the 20J/HDPE composite exhibited an increase of 50% in strength and 193% in the value of E compared to the pure HDPE studied by Mendes et al. [67]. Similarly, the 20J/GNP/HDPE single nanocomposite showed significant improvements, with a 45% increase in strength and a remarkable 262% increase in the E compared to pure HDPE. Regarding the 10J/10A/GNP/HDPE, it presented an increase superior to 900% in tensile strength, as well as 256% in the E. In addition, there was a reduction of 89% in ductility when compared with pure HDPE. These results highlight the potential of using a combination of synthetic and natural fibers to enhance the mechanical properties of thermoplastic matrix hybrid nanocomposites.

### 3.4. Izod Impact Test

Figure 21 depicts the average Izod impact absorption capacities of the 20J/HDPE composite and 20J/GNP/HDPE single nanocomposite. One can notice that the energy absorbed by the 20J/GNP/HDPE (349.70 ± 12.03 J/m) is significantly higher than that of the 20J/HDPE (308.00 ± 17.43 J/m), representing an increase of 14% in absorbed energy. These values are comparable to those of polymer composites reinforced with other natural fibers [55,67,68].

Although the degree of crystallinity decreases the absorption energy capability of the matrix, the 20J/GNP/HDPE nanocomposite presented a higher impact resistance. This phenomenon could be attributed to the size of the GNP particles, which were larger than 25 μm, and were effective in absorbing the energy from the impact.

However, when evaluating the properties of the jute/aramid/GNP/HDPE hybrid nanocomposites, significantly larger values in the energy absorbed upon Izod impact is observed, as evidenced by Figure 22. The ANOVA reveals that the calculated F value of 18.66 exceeds the critical F value of 3.06, indicating that the mean values are significantly different with a 95% confidence level. Hence, Tukey’s test was employed to identify the groups that exhibit significant differences in absorbed energy values.

As per the Tukey test, in Table 9, with an msd of 91.88, all hybrid nanocomposites exhibited a significant difference compared to the 20J/HDPE composite. Notably, the 10J/10A/GNP/HDPE hybrid nanocomposite demonstrated the highest value of absorbed energy at 583.52 ± 48.59 J/m, as indicated in Table 10 and Figure 22. This represents a substantial increase of 89% compared to the 20J/HDPE composite, and a 20% increase compared to the 20A/HDPE composite.

The absorbed energy value of the 10J/10A/GNP/HDPE hybrid nanocomposite surpasses that of several other polymeric composites reported in the literature, Table 10. Thus, with an increase in the number of aramid fabric layers, the absorbed energy upon Izod impact decreases after reaching 10 layers. This phenomenon is related to the fracture behavior of the composites, where at lower concentrations of aramid fibers, more fractures occur in the natural fibers and the HDPE matrix. On the other hand, with a higher aramid concentration, fewer fractures in the reinforcement are observed, resulting in lower energy absorption [64]. This mechanism is supported by electron microscopy analysis of the specimens after the Izod impact test, shown in Figure 23.

The fracture mechanism of the hybrid nanocomposites with both natural and synthetic fibers, as shown in Figure 23, reveals more delamination and less fiber rupture, particularly in the aramid fibers. Additionally, defibrillation of the aramid fibers and a complete rupture of the jute fibers can be observed.

## 4. Conclusions

The Raman spectroscopy revealed the characteristic bands of the plain HDPE matrix, as well as the presence of the D and G bands after the incorporation of the 0.10 wt.% GNP of the GNP/HDPE nanocomposite.The presence of GNP particles did not enhance the crystallization process in the GNP/HDPE nanocomposites, leading to lower crystallization rates at higher crystalline fractions. However, the nucleation mechanism and crystal growth geometries were consistent with those found in the literature.The Friedman’s differential isoconversional method confirmed the nucleation effect of GNP on the initial stages of HDPE crystallization by reducing the activation energy for lower crystalline fractions (*X_T_* ≤ 30%), which is the range where the maximum crystallization rate is achieved. However, beyond 30% of crystallization, the GNP appeared to increase the complexity of the crystal-growing process.In comparison with the 20/JHDPE, the 20J/GNP/HDPE single nanocomposite displays E and T values higher than 20 and 40%, respectively. In addition, the 20J/GNP/HDPE showed significant improvements, with a 45% increase in strength as well as 262% in the value of E in comparison to pure HDPE.The hybrid nanocomposites demonstrated remarkable improvements in mechanical properties. Specifically, the 10J/10A/GNP/HDPE displayed an impressive 2433% increase in mechanical toughness compared to the 20J/GNP/HDPE single nanocomposite. Additionally, it exhibited a 591% increase in tensile strength and a 462% reduction in ductility. These findings underscore the potential of utilizing a combination of synthetic and natural fibers to bolster the mechanical properties of thermoplastic matrix nanocomposites.The addition of 0.10 wt.% GNP to the 20J/GNP/HDPE nanocomposite significantly increased impact strength by 14% compared to the pure 20J/HDPE composite, demonstrating a reinforcing effect of the GNP. Similarly, the 10J/10A/GNP/HDPE hybrid nanocomposite exhibited the highest impact strength, with a 90% increase compared to 20J/HDPE and a 67% increase compared to 20J/GNP/HDPE.In a 20J/HDPE composite, the primary fracture mechanism was shear band deformation of the HDPE matrix. However, with the addition of 0.10 wt.% GNP, the fracture mechanism of the nanocomposite changed to a combination of fibrillation, tearing, and voids. The jute/aramid/GNP/HDPE hybrid nanocomposites showed a complex fracture behavior due to the use of two different fabrics. This behavior included delamination, fiber rupture, matrix rupture, fibrillation, and tearing of the GNP/HDPE. Higher aramid layers resulted in more delamination and less fiber rupture. Overall, the results of these novel hybrid aramid/jute GNP/HDPE matrix nanocomposites revealed promising materials for diversified engineering applications.

## Figures and Tables

**Figure 1 polymers-15-02460-f001:**
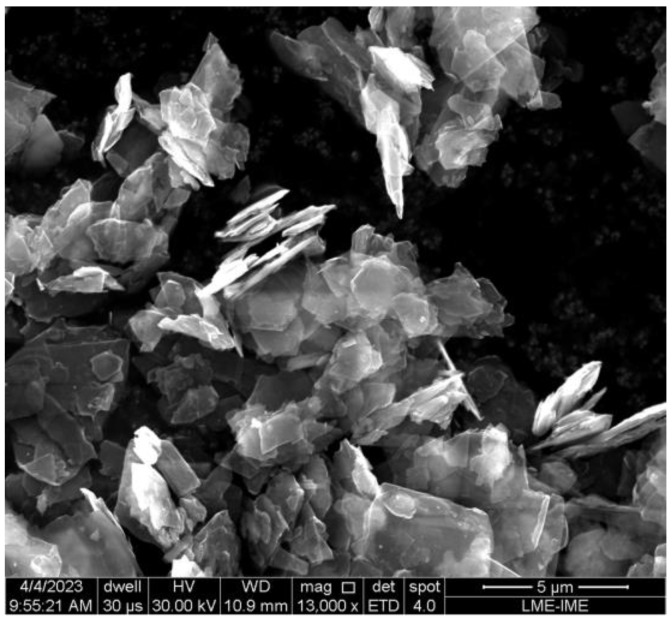
SEM view of the GNP size and morphology used in this work.

**Figure 2 polymers-15-02460-f002:**
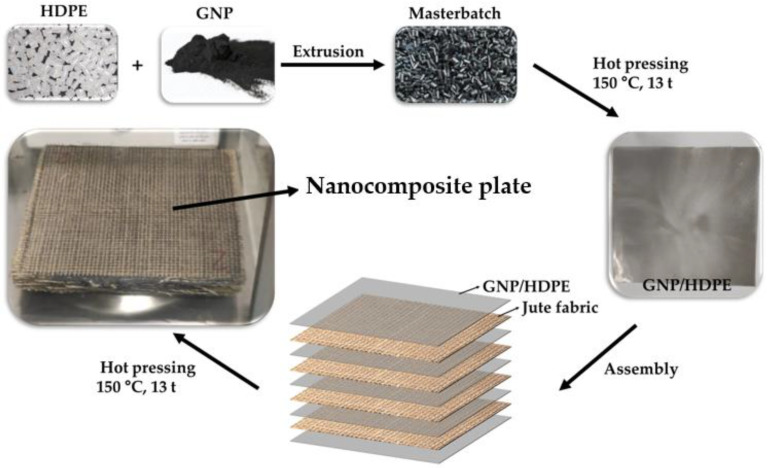
Schematic illustration of the fabrication of the nanocomposite plates.

**Figure 3 polymers-15-02460-f003:**
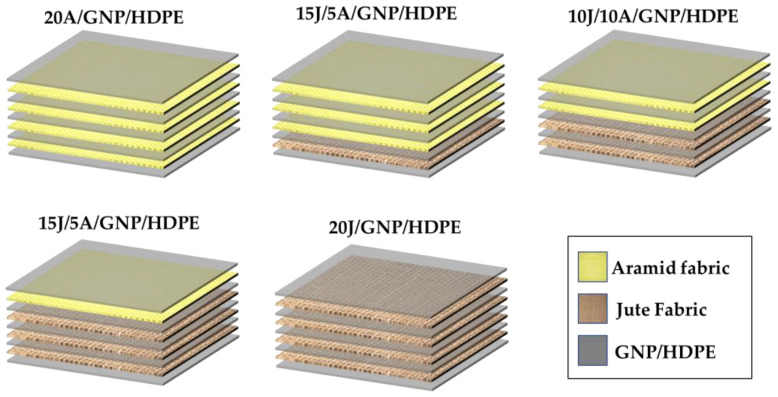
Scheme of the fabrication of the hybrid jute and aramid fabrics-reinforced GNP/HDPE nanocomposites.

**Figure 4 polymers-15-02460-f004:**
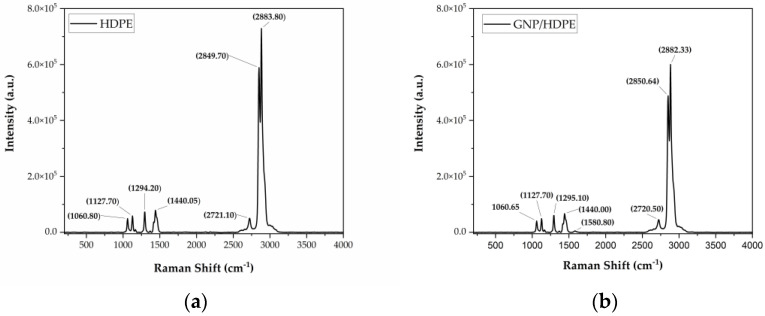
RAMAN spectra of the (**a**) HDPE and (**b**) GNP/HDPE films.

**Figure 5 polymers-15-02460-f005:**
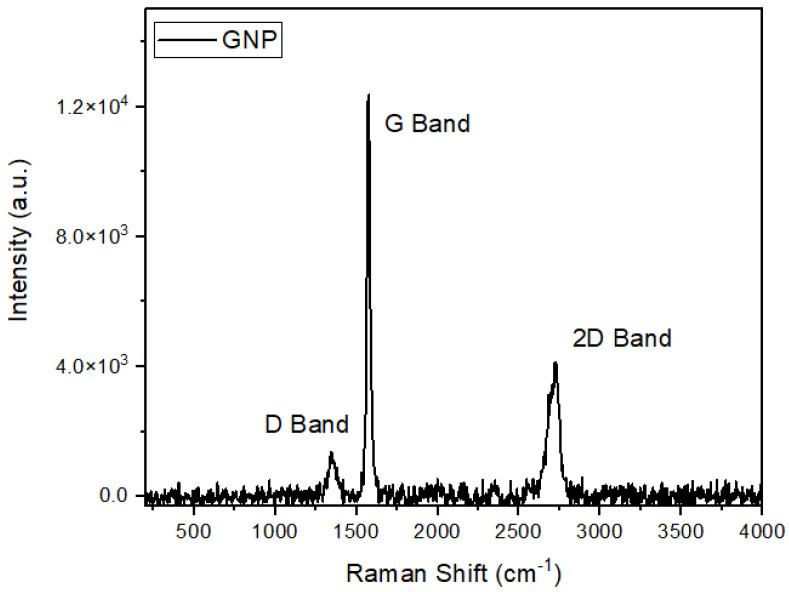
RAMAN spectrum of GNPs.

**Figure 6 polymers-15-02460-f006:**
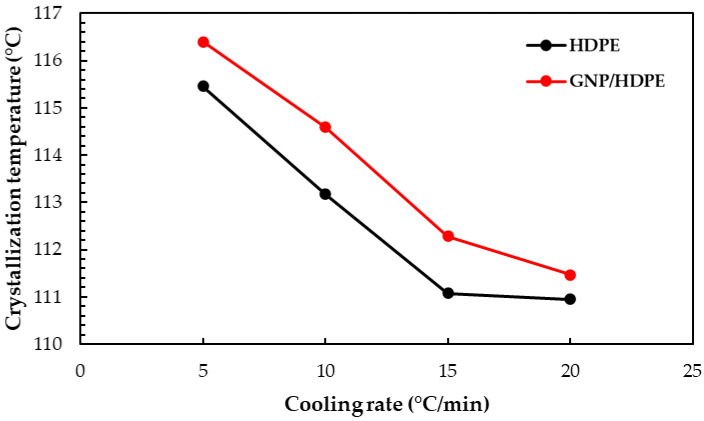
Crystallization temperature of the neat HDPE and GNP/HDPE nanocomposite.

**Figure 7 polymers-15-02460-f007:**
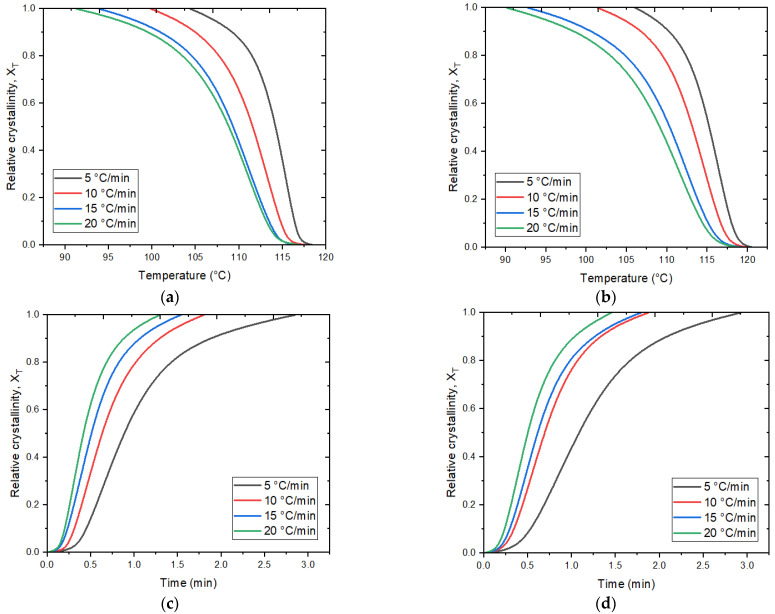
Graphs of relative crystallinity for non-isothermal crystallization at cooling rates from 5 to 20 °C/min as a function of temperature (**a**) HDPE and (**b**) GNP/HDPE, and as a function of time (**c**) HDPE and (**d**) GNP/HDPE.

**Figure 8 polymers-15-02460-f008:**
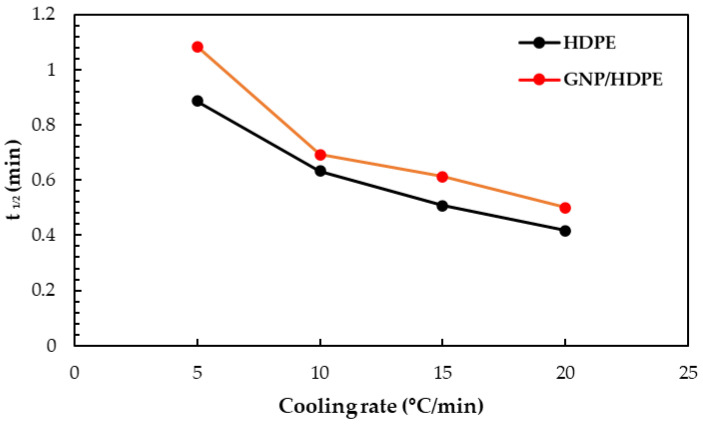
Half-time of crystallization for neat HDPE and GNP/HDPE nanocomposite.

**Figure 9 polymers-15-02460-f009:**
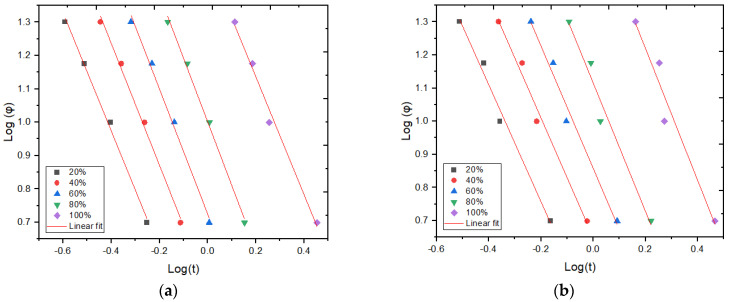
Log(φ) versus Log(t) graphs for non-isothermal crystallization of the nanocomposites: (**a**) HDPE and (**b**) GNP/HDPE.

**Figure 10 polymers-15-02460-f010:**
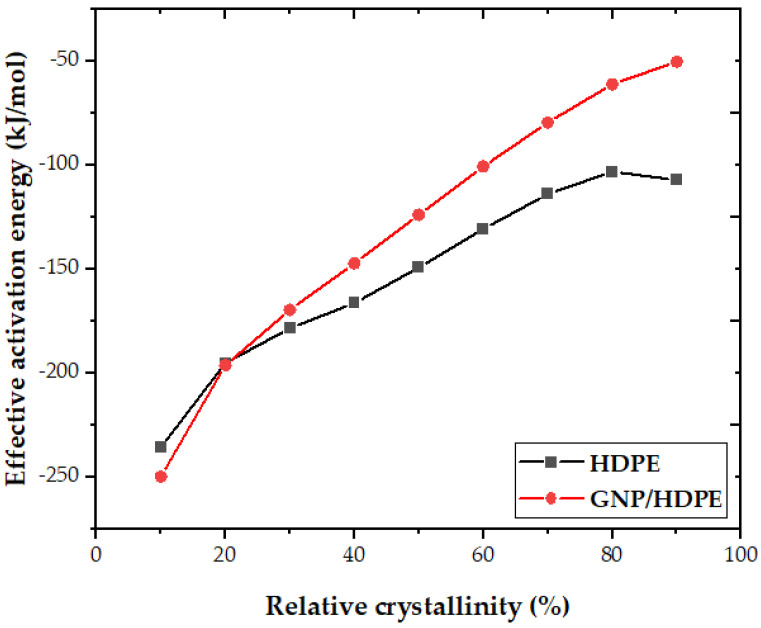
The influence of crystallinity on the effective activation energy of neat HDPE and GNP/HDPE nanocomposites.

**Figure 11 polymers-15-02460-f011:**
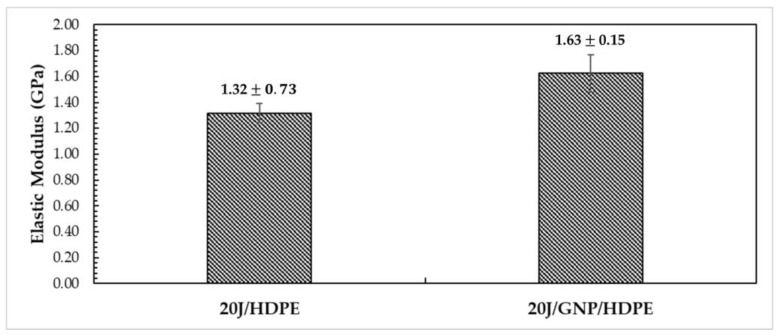
Elastic modulus of jute/HDPE composite and jute/GNP/HDPE single nanocomposites.

**Figure 12 polymers-15-02460-f012:**
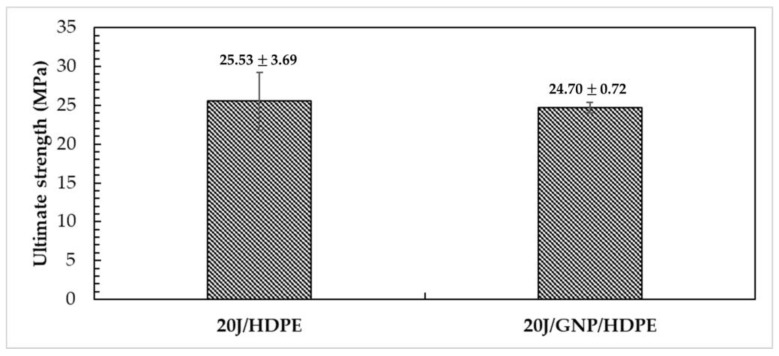
Tensile strength of jute/HDPE composite and jute/GNP/HDPE single nanocomposite.

**Figure 13 polymers-15-02460-f013:**
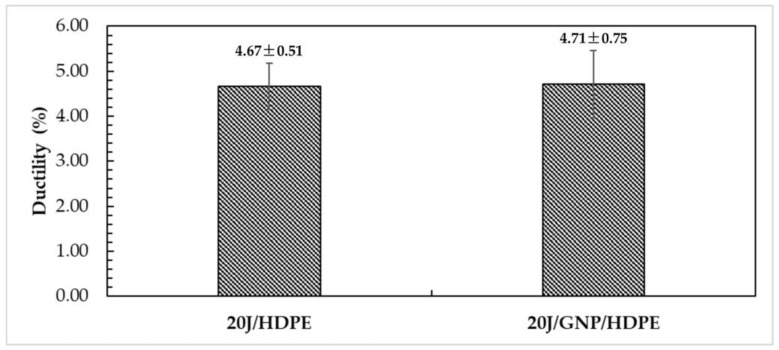
Ductility of jute/HDPE composite and jute/GNP/HDPE single nanocomposite.

**Figure 14 polymers-15-02460-f014:**
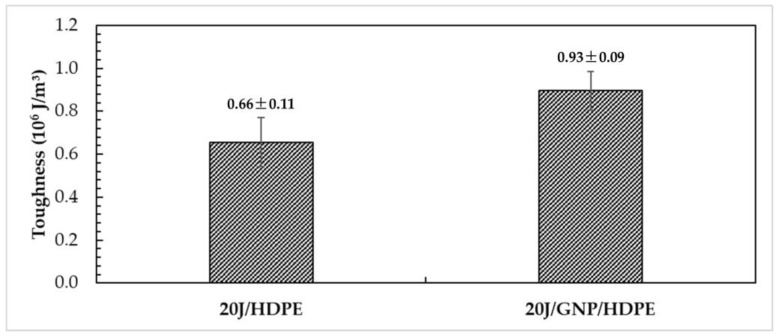
Toughness of jute/HDPE composite and jute/GNP/HDPE single nanocomposite.

**Figure 15 polymers-15-02460-f015:**
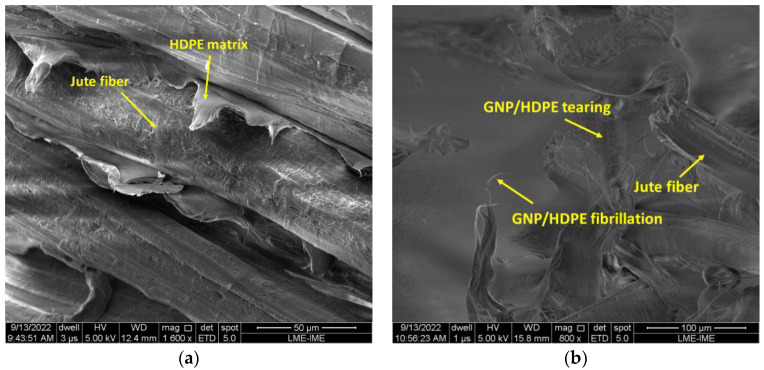
Fracture surfaces of composites (**a**) 20J/HDPE and (**b**) 20J/GNP/HDPE.

**Figure 16 polymers-15-02460-f016:**
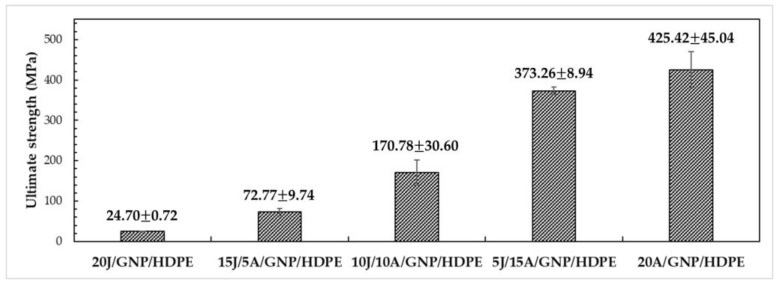
Tensile strength of single and hybrid nanocomposites.

**Figure 17 polymers-15-02460-f017:**
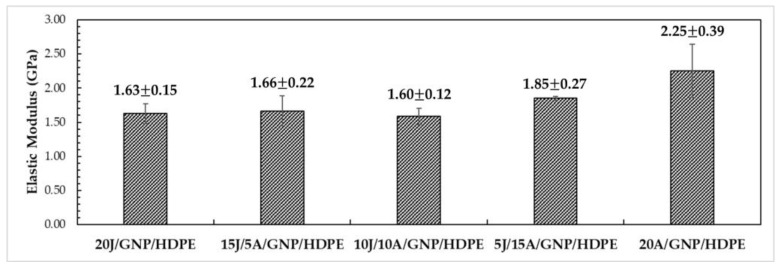
Elastic modulus of single and hybrid nanocomposites.

**Figure 18 polymers-15-02460-f018:**
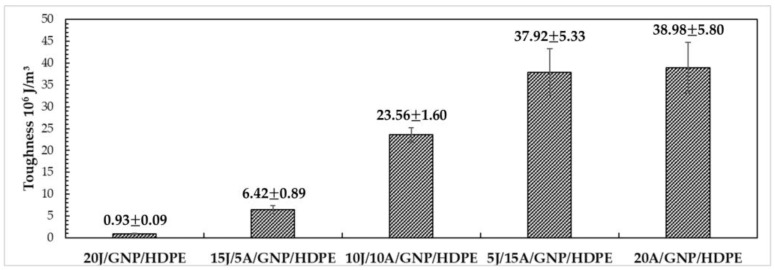
Toughness of single and hybrid nanocomposites.

**Figure 19 polymers-15-02460-f019:**
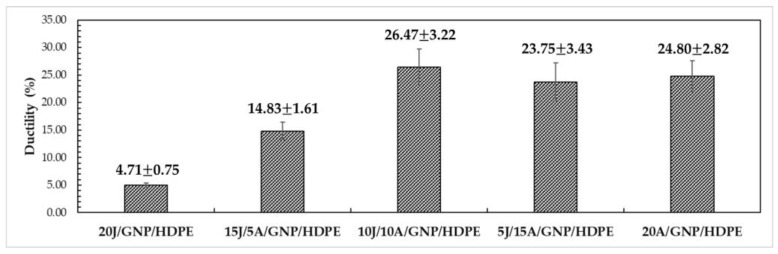
Ductility of single and hybrid nanocomposites.

**Figure 21 polymers-15-02460-f021:**
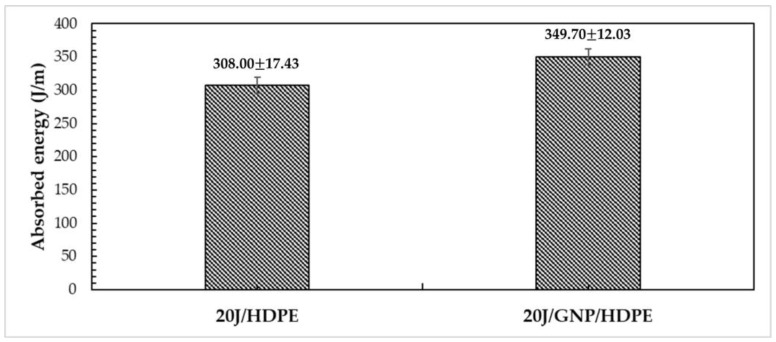
Energy absorbed by jute/HDPE composite as well as jute/GNP/HDPE nanocomposites and through Izod impact.

**Figure 22 polymers-15-02460-f022:**
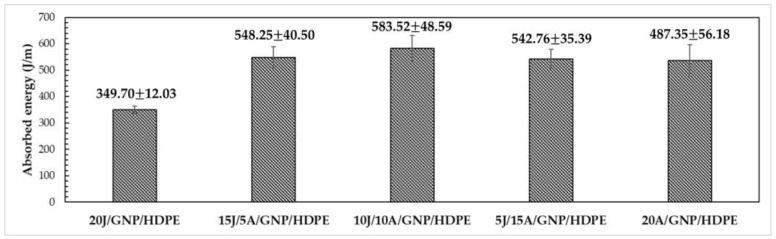
Energy absorbed by single and hybrid nanocomposites through Izod impact.

**Figure 23 polymers-15-02460-f023:**
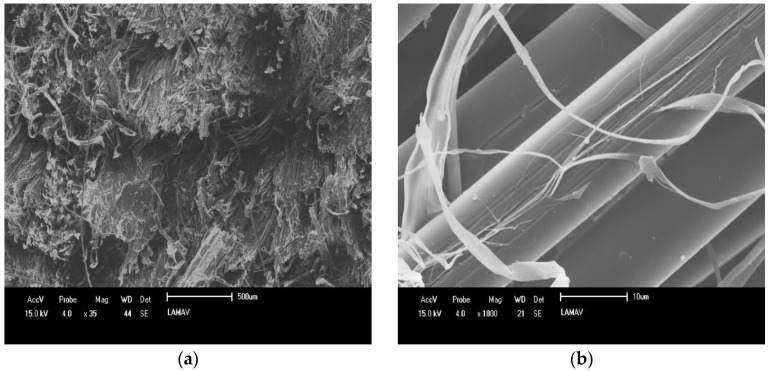
Fracture surface of group 2 composites: (**a**) 10J/10A/GNP/HDPE and (**b**) 20A/GNP/HDPE.

**Table 1 polymers-15-02460-t001:** Nanocomposites configurations and associated nomenclatures.

Configuration	Nomenclature	Volume Fraction of Fabrics	Percentage of Layers of Aramid
Zero layers of jute/20 layers of aramid/GNP-functionalized HDPE matrix (single nanocomposite)	20A/GNP/HDPE	0 vol% Jute + 50 vol% Aramid	100%
5 layers of jute/15 layers of aramid/GNP-functionalized HDPE matrix (hybrid nanocomposite)	5J/15A/GNP/HDPE	12.5 vol% Jute + 37.5 vol% Aramid	75%
10 layers of jute/10 layers of aramid/GNP-functionalized HDPE matrix (hybrid nanocomposite)	10J/10A/GNP/HDPE	25 vol% Jute + 25 vol% Aramid	50%
15 layers of jute/5 layers of aramid/GNP-functionalized HDPE matrix (hybrid nanocomposite)	15J/5A/GNP/HDPE	37.5 vol% Jute + 12.5 vol% Aramid	25%
20 layers of jute/zero layers of aramid/GNP-functionalized HDPE matrix (single nanocomposite)	20J/GNP/HDPE	50 vol% Jute + 0 vol% Aramid	0%
20 layers of jute/zero layers of aramid/plain HDPE matrix (single composite)	20J/HDPE	50 vol% Jute + 0 vol% Aramid	0%

**Table 2 polymers-15-02460-t002:** Crystallization and crystalline melting temperatures of composites for each heating rate.

Crystallization Temperature	HDPE	GNP/HDPE
5 °C/min	115.46	116.40
10 °C/min	113.17	114.59
15 °C/min	111.08	112.28
20 °C/min	110.95	111.47
Average	112.67	113.69
**Melting Temperature**		
5 °C/min	133.30	134.50
10 °C/min	133.90	133.40
15 °C/min	131.80	133.80
20 °C/min	132.40	133.30
Average	132.85	133.75

**Table 3 polymers-15-02460-t003:** Kinetic parameters of crystallization based on the method of Liu et al. [52] for neat HDPE and GNP/HDPE nanocomposites, under non-isothermal conditions.

	*X_T_* (%)	*α*	*Log*[*F*(*T*)]	R^2^
**HDPE**	20	1.8	0.3	0.99683
	40	1.8	0.5	0.99533
	60	1.9	0.7	0.99425
	80	1.9	1.0	0.99644
	100	1.8	1.5	0.99042
**GNP/HDPE**	20	1.8	0.4	0.98809
	40	1.8	0.6	0.97888
	60	1.8	0.9	0.98201
	80	1.9	1.1	0.97077
	100	2.0	1.6	0.95336

**Table 4 polymers-15-02460-t004:** Tukey test for tensile strength values of single and hybrid nanocomposites.

	20J/GNP/HDPE	15J/5A/GNP/HDPE	10J/10A/GNP/HDPE	5J/15A/GNP/HDPE	20A/GNP/HDPE
**20J/GNP/HDPE**	0.00	47.91	145.93	348.41	400.57
**15J/5A/GNP/HDPE**	47.91	0.00	98.02	300.50	352.66
**10J/10A/GNP/HDPE**	145.93	98.02	0.00	202.49	254.65
**5J/15A/GNP/HDPE**	348.41	300.50	202.49	0.00	52.16
**20A/GNP/HDPE**	400.57	352.66	254.65	52.16	0.00

**Table 5 polymers-15-02460-t005:** Tukey test for the E of single and hybrid nanocomposites.

	20J/GNP/HDPE	15J/5A/GNP/HDPE	10J/10A/GNP/HDPE	5J/15A/GNP/HDPE	20A/GNP/HDPE
**20J/GNP/HDPE**	0.00	39.20	40.20	224.97	628.03
**15J/5A/GNP/HDPE**	39.20	0.00	79.40	185.77	588.83
**10J/10A/GNP/HDPE**	40.20	79.40	0.00	265.17	668.23
**5J/15A/GNP/HDPE**	224.97	185.77	265.17	0.00	403.06
**20A/GNP/HDPE**	628.03	588.83	668.23	403.06	0.00

**Table 6 polymers-15-02460-t006:** Tukey test for mechanical toughness values of single and hybrid nanocomposites.

	20J/GNP/HDPE	15J/5A/GNP/HDPE	10J/10A/GNP/HDPE	5J/15A/GNP/HDPE	20A/GNP/HDPE
**20J/GNP/HDPE**	0.00	5.49	22.63	34.71	38.04
**15J/5A/GNP/HDPE**	5.49	0.00	17.14	29.21	32.55
**10J/10A/GNP/HDPE**	22.63	17.14	0.00	12.07	15.41
**5J/15A/GNP/HDPE**	34.71	29.21	12.07	0.00	3.34
**20A/GNP/HDPE**	38.04	32.55	15.41	3.34	0.00

**Table 7 polymers-15-02460-t007:** Tukey test for ductility values of single and hybrid nanocomposites.

	20J/GNP/HDPE	15J/5A/GNP/HDPE	10J/10A/GNP/HDPE	5J/15A/GNP/HDPE	20A/GNP/HDPE
**20J/GNP/HDPE**	0.00	10.12	21.76	19.04	20.09
**15J/5A/GNP/HDPE**	10.12	0.00	11.63	8.91	9.97
**10J/10A/GNP/HDPE**	21.76	11.63	0.00	2.72	1.67
**5J/15A/GNP/HDPE**	19.04	8.91	2.72	0.00	1.06
**20A/GNP/HDPE**	20.09	9.97	1.67	1.06	0.00

**Table 9 polymers-15-02460-t009:** Tukey test for absorbed energy values of single and hybrid nanocomposites.

	20J/GNP/HDPE	15J/5A/GNP/HDPE	10J/10A/GNP/HDPE	5J/15A/GNP/HDPE	20A/GNP/HDPE
**20J/GNP/HDPE**	0.00	194.37	229.64	188.89	183.27
**15J/5A/GNP/HDPE**	194.37	0.00	35.27	5.48	11.10
**10J/10A/GNP/HDPE**	229.64	35.27	0.00	40.76	46.37
**5J/15A/GNP/HDPE**	188.89	5.48	40.76	0.00	5.61
**20A/GNP/HDPE**	183.27	11.10	46.37	5.61	0.00

**Table 10 polymers-15-02460-t010:** Summary of the absorbed energy discussed in the present work in comparison with other works in the literature.

Composite	Impact Resistance (J/m)	Reference
20J/HDPE	308.00 ± 17.43	PW
20J/GNP/HDPE	349.70 ± 12.03	
15J/5A/GNP/HDPE	548.25 ± 40.50	
10J/10A/GNP/HDPE	583.52 ± 48.59	
5J/15A/GNP/HDPE	542.76 ± 35.39	
20A/GNP/HDPE	487.35 ± 56.18	
HDPE	373.65 ± 30.00	[55]
GNP/HDPE	213.52 ± 20.00	[67]
CaCO_3_/FS/HDPE	90.00 ± 5.67	[68]

PW—Present work.

## Data Availability

All data underlying the results are available as part of the article and no additional source data are required.

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
