# Peer review of "Mechanical Properties Optimization of Hybrid Aramid and Jute Fabrics-Reinforced Graphene Nanoplatelets in Functionalized HDPE Matrix Nanocomposites"

_polymers, 2023, doi:10.3390/polym15112460_

Round 1
Reviewer 1 Report
Though the proposed work is unique, the paper is prepared just as a scratch and requires considerable improvement before recommending the paper for publication. The following comments are to be addressed carefully while preparing the revised manuscript.
1. The paper contains a lot of grammatical errors and sentence revisions. The authors should avoid forming "compound sentences" as it will be difficult for the readers to understand the findings. Hence, they are to be re-phrased in to simple sentences. Moreover, the entire paper is to be thoroughly checked and all the grammatical errors are to be rectified in the revised submission.
2. Abstract of the paper is to be made concise by removing redundant sentences and presenting within the word limit of 150 words. At present, it looks like an abstract of thesis or report and and to be revised as per the journal format.
3. The reviewer feels that the depth of introduction section is to be improved significantly. A number of recent papers related to topic is missed and to be added to the revised submission. This is a major set back for the revised paper.
4. Section 2 presents just minimum details without any proper justification.
5. The fabrication of nano-composites in the present work is to be elaborated for the benefit of the readers.
6. DSC curves (both exothermic and endothermic) obtained for the present composite is to be added. Moreover, clarify what is the need for performing DSC in the present work.
7. Observations made from the Raman spectroscopy is to be added to the revised submission.
8. Conclusions also require a careful re-writing to remove the redundant data and present only important findings from the present work.
8.
The paper contains a lot of grammatical errors and sentence revisions. The authors should avoid forming "compound sentences" as it will be difficult for the readers to understand the findings. Hence, they are to be re-phrased in to simple sentences. Moreover, the entire paper is to be thoroughly checked and all the grammatical errors are to be rectified in the revised submission.
Author Response
Response to the Reviewer
The authors would like to thank the Reviewer for the valuable comments and suggestions on the structure and scientific aspects that contribute to improve the manuscript. Amendments are provided accordingly. Responses to each comment are listed below and all modifications/additions were marked as Track changes in the revised version of the manuscript.
General comment: Though the proposed work is unique, the paper is prepared just as a scratch and requires considerable improvement before recommending the paper for publication. The following comments are to be addressed while preparing the revised manuscript.
Response: The author would like to thank the Reviewer general comments, and the improvements were done in the revised manuscript carefully as requested by the Reviewer.
Comment (1): The paper contains a lot of grammatical errors and sentence revisions. The authors should avoid forming "compound sentences" as it will be difficult for the readers to understand the findings. Hence, they are to be re-phrased in to simple sentences. Moreover, the entire paper is to be thoroughly checked and all the grammatical errors are to be rectified in the revised submission.
Response: The authors understand the important point raised by the reviewer and English language was thoroughly revised with the assistance of an expert to enhance the readability of the manuscript. We hope that this new version is suitable for publication in Polymers.
Comment (2): Abstract of the paper is to be made concise by removing redundant sentences and presenting within the word limit of 150 words. At present, it looks like an abstract of thesis or report and and to be revised as per the journal format.
Response: Complied. The abstract was modified and now is presented with less than 150 words against the 187 words observed in the previous version.
Comment (3): The reviewer feels that the depth of introduction section is to be improved significantly. A number of recent papers related to topic is missed and to be added to the revised submission. This is a major set back for the revised paper.
Response: The authors understand the Reviewer’s comment. In the new version some up-to-date references were added in the introduction section.
Comment (4): Section 2 presents just minimum details without any proper justification.
Response: The Reviewer is right and, more information with proper justification is included in section 2.
Comment (5): The fabrication of nano-composites in the present work is to be elaborated for the benefit of the readers.
Response: As recommended, the fabrication of nanocomposites is now elaborated.
Comment (6): DSC curves (both exothermic and endothermic) obtained for the present composite is to be added. Moreover, clarify what is the need for performing DSC in the present work.
Response: Complied, in the new version of the manuscript, the importance of the DSC analysis is included in the text. In addition, both exothermic and endothermic curves are included as supplementary materials.
Comment (7): Observations made from the Raman spectroscopy is to be added to the revised submission.
Response: Complied, the Raman peaks are included in the figures, as well as tables containing all the observed peaks are included as supplementary materials.
Comment (8): Conclusions also require a careful re-writing to remove the redundant data and present only important findings from the present work.
Response: The authors agree with the Reviewer, and the conclusion is now shortened in the revised version of the manuscript, highlighting the most important results.
Comment (9): The paper contains a lot of grammatical errors and sentence revisions. The authors should avoid forming "compound sentences" as it will be difficult for the readers to understand the findings. Hence, they are to be re-phrased in to simple sentences. Moreover, the entire paper is to be thoroughly checked and all the grammatical errors are to be rectified in the revised submission.
Response: The authors would like to thank the Reviewer for the important suggestions, and once again we would like to point out that English language was thoroughly revised with the assistance of an expert in this new revised version.
Reviewer 2 Report
It is an original paper dealing with optimizing tensile and impact resistance properties of novel jute/aramid/high-density polyethylene (HDPE) hybrid nanocomposites by adding graphene nano-platelets. The paper is well-written, and there are some minor comments to make it more beneficial for the readers.
· The abstract must be rewritten, shorten the initial sentences, and add a couple of sentences to highlight the main achievements of the work.
· In the last paragraph of the introduction, the novelty of the work must be described.
· As the main objective of the manuscript is an investigation of nanoparticles on the mechanical properties of composites, it is recommended to add more research regarding adding these particles to different materials (may it is worth it to add a table including the used polymer, fibre, optimized nano-particles content, …) such as the below ones.
Effects of nanoparticles on nanocomposites mode I and II fracture: A critical review
Impedance analysis for condition monitoring of single lap CNT-epoxy adhesive joint
Trapezoidal traction–separation laws in mode II fracture in nano-composite and nano-adhesive joints
· The investigation has been done on only 0.1%wt GNP. It is necessary to compare the results for different percentages of GNPs.
Author Response
Response to the Reviewer
The authors would like to thank the Reviewer for the valuable comments and suggestions on the structure and scientific aspects that contribute to improve the manuscript. Amendments are provided accordingly. Responses to each comment are listed below and all modifications/additions were marked as Track changes in the revised version of the manuscript.
General comment: It is an original paper dealing with optimizing tensile and impact resistance properties of novel jute/aramid/high-density polyethylene (HDPE) hybrid nanocomposites by adding graphene nano-platelets. The paper is well-written, and there are some minor comments to make it more beneficial for the readers.
Response: The authors are grateful for the insightful comments and suggestions provided by the reviewer.
Comment (1): The abstract must be rewritten, shorten the initial sentences, and add a couple of sentences to highlight the main achievements of the work.
Response: Complied, in the revised version of the manuscript a short abstract is now provided. With indication of the main achievements of our work.
Comment (2): In the last paragraph of the introduction, the novelty of the work must be described.
Response: The authors agree with the reviewer, and the novelty of the research is now included in the introduction.
Comment (3): As the main objective of the manuscript is an investigation of nanoparticles on the mechanical properties of composites, it is recommended to add more research regarding adding these particles to different materials (may it is worth it to add a table including the used polymer, fibre, optimized nano-particles content, …) such as the below ones.
“Effects of nanoparticles on nanocomposites mode I and II fracture: A critical review”
“Impedance analysis for condition monitoring of single lap CNT-epoxy adhesive joint”
“Trapezoidal traction–separation laws in mode II fracture in nano-composite and nano-adhesive joints”
Response: The authors would like to thank the Reviewer for the important background paper suggested for the improvement of the manuscript. In the revised version these references are now included.
Comment (4): The investigation has been done on only 0.1%wt GNP. It is necessary to compare the results for different percentages of GNPs.
Response: The Reviewer has raised an important point about nanocomposites. Indeed, the investigation is more interesting when comparing different amounts of GNPs. However, we could infer from background papers, that lower concentrations of nanoparticles are more effective, because of the agglomeration process that occurs in higher amounts. Thus, in this work, we focused on only one percentage of nanofiller, which in fact is considered one of the most used in the literature. Furthermore, our current investigation on other GNP amounts is included in a coming paper that is being considered ongoing for publication.
Round 2
Reviewer 1 Report
Comments were addressed.